# Hand Grip Force–Time Curve Indicators Evaluated by Dynamometer: A Systematic Review

**DOI:** 10.3390/nu16121951

**Published:** 2024-06-19

**Authors:** Tânia Silva-Santos, Rita S. Guerra, Rui Valdiviesso, Teresa F. Amaral

**Affiliations:** 1LAETA-INEGI/FEUP, Associated Laboratory of Energy, Transports and Aerospace, Institute of Science and Innovation in Mechanical and Industrial Engineering, Faculty of Engineering, University of Porto, 4200-465 Porto, Portugal; taniiasilvasantos@gmail.com (T.S.-S.); ritaguerra@ufp.edu.pt (R.S.G.); 2FP-I3ID, FP-BHS, Faculty of Health Sciences, University Fernando Pessoa, 4200-150 Porto, Portugal; 3FCNAUP, Faculty of Nutrition and Food Sciences, University of Porto, 4150-180 Porto, Portugal; ruivaldiviesso@fcna.up.pt; 4CINTESIS@RISE, MEDCIDS, Faculty of Medicine, University of Porto, 4200-450 Porto, Portugal

**Keywords:** force–time curve, hand grip strength, dynamometer

## Abstract

Background: Handgrip strength (HGS) is an indicator of muscular strength, used in the diagnosis of sarcopenia, undernutrition, and physical frailty as well as recovery. Typically, the maximum HGS value is used; however, recent evidence suggests the exploration of new indicators provided based on the force–time curve to achieve a more comprehensive assessment of muscle function. Therefore, the objective was to identify indicators of the HGS profile beyond maximum HGS, based on force–time curves, and to systematize knowledge about their applications to various types of samples, health issues, and physical performance. Methods: A systematic review was performed including studies whose participants’ HGS was assessed with a digital or adapted dynamometer. The outcome measures were HGS profile indicators calculated from the force–time curve. Results: a total of 15 studies were included, and the following indicators were identified: grip fatigue, fatigability index, fatigue rate, fatigue resistance, time to 80% maximal voluntary contraction, plateau coefficient of variability, time to maximum value, T-90%, release rate, power factor, grip work, average integrated area, endurance, cycle duration, time between cycles, maximum and minimum force–velocity, rate of grip force, final force, inflection point, integrated area, submaximal control, and response time. Conclusions: Various indicators based on the force–time curve can be assessed through digital or adapted dynamometers. Future research should analyze these indicators to understand their implications for muscle function assessment, to standardize evaluation procedures, to identify clinically relevant measures, and to clarify their implications in clinical practice.

## 1. Introduction

Hand grip strength (HGS) is an easy-to-perform, reliable, and low-cost indicator of muscular strength, which is related to several adverse health events and has been used in several clinical areas [1,2,3,4,5,6]. Many studies have reported that HGS decreases from middle age [7] and is associated with upper limb impairment [8,9,10], chronic fatigue, developmental disabilities [11], muscular dystrophy [2], new-onset cardiometabolic diseases [12,13,14,15], falls [16,17], hospitalizations [18], morbidity and mortality from the disease [19,20,21], and low health-related quality of life [22,23]. It is a diagnostic tool to identify sarcopenia, undernutrition, physical frailty, and recovery from these diseases and processes [24,25,26]. Moreover, HGS is a commonly used parameter to assess physical performance and fitness [2]. The assessment of maximum-strength muscle contractions depends on the recruitment of the motor unit [27]. The neuromuscular system helps control fine coordinated movements, especially in the hands, including the ability to control muscle force over time [28].

Objective measures of HGS can be obtained using a dynamometer. However, methods for assessing HGS vary considerably regarding the used dynamometer, the measurement protocol, which hand is used, or if both hands are used, and all these factors may affect HGS values [24,29,30]. The maximum value of the isometric strength exertion of the hand grip is widely used in the assessments of HGS [6,31,32]. Despite the robust health insights offered by standalone maximum HGS measurements, their clinical significance in guiding strategies for preventing and treating weakness remains restricted due to the unclear etiology of muscle weakness. Other characteristics of muscle function in addition to strength capacity may be relevant because muscle function is multivariable [33]. Hence, maximal HGS serves as a strong but still partial indicator of muscle function [24].

This limitation to maximal HGS values led researchers to explore the possibilities of the HGS force–time curve that can be assessed through adapted hand dynamometers or digital dynamometers and provides a more detailed description of force development and maintenance. Digital hand grip dynamometry offers innovative prospects for achieving a more comprehensive evaluation of muscle function, extending beyond mere strength capacity, all while preserving procedural simplicity. The HGS force–time curve is a graphical representation that displays the force exerted by contracting muscles (*y*-axis) over a predefined time interval (*x*-axis) [34]. The typical force–time curve exhibits a rapid initial increase in force (force generation phase), followed by a smooth peak curve period (plateau phase) and a subsequent gradual decrease in force over time (force decay or fatigue phase) [35].

Digital hand grip dynamometry enables the assessment of novel force–time curve indicators, such as the rate of force development that is independent of maximum grip strength [25,36] or grip fatigue [37,38,39], fatigability index [40,41], fatigue rate [39], and fatigue resistance [42,43,44]. The use of such indicators to explore additional aspects of muscle function can help in identifying disease-related motor changes and revealing mechanisms responsible for age-related impairments, which are not detectable when based only on maximal HGS assessments, such as fatigue [45,46] or the submaximal control indicator [25]. Furthermore, it might enable healthcare professionals to pinpoint the risk of age-related illness and disability, thereby enhancing the accuracy of referrals [25,36]. 

Several HGS indicators derived from the force–time curve have been described in the literature over the last few decades by authors from diverse scientific areas. However, this relevant information has been found in a dispersed manner. To the best of our knowledge, a review of all indicators of HGS based on the force–time curve evaluated by a dynamometer has not yet been carried out and remains of major relevance. The objective of this systematic review was to identify indicators of the HGS measurement profile based on the force–time curve and to systematize knowledge about their applications to various types of samples, health issues, and physical performance.

## 2. Materials and Methods

This systematic review was written according to the Preferred Reporting Items for Systematic Reviews and Meta-Analyzes (PRISMA) [47].

### 2.1. Eligibility Criteria 

Inclusion criteria were studies published in English or Portuguese, randomized clinical trials, or other experimental, cohort, and cross-sectional studies. Participants were adults (>18 years old) and residents of any region of the world. All populations were included regardless of health status.

The type of exposure evaluated was muscle strength assessed by gripping hands with a dynamometer, and the outcome measures were HGS profile indicators, obtained through the force–time curve, in addition to maximum strength. All studies that only assessed maximum strength and studies that did not describe the strength profile indicator were excluded.

### 2.2. Information Sources 

The studies were identified by searching on 25 August 2023, in electronic databases and scanning reference lists of the articles included in this article and in systematic reviews or protocols that emerged in the search for data. As the selection of articles took more than two months, a new search was carried out on November 13 to find any new articles and thus to ensure that the review included all recent evidence. The databases searched were PubMed and Scopus.

### 2.3. Search Strategy

The search was performed by one author and included two categories: “hand grip strength” terms and “dynamometer” terms. To achieve the keywords, the Mesh term “hand strength” was used. Furthermore, synonyms that were not included in the Mesh term were verified and reviewed in articles on the topic. The term “dynamometer” was used because it is the tool that evaluates our outcome. No synonyms were found for “dynamometer”.

In PubMed, the search key was “(hand strength[Mesh]) OR (grip strength) OR (explosive AND (force OR grip OR strength)) OR (force-time)) AND (dynamometer)”. In Scopus, the search key was “(hand strength) OR (grip strength) OR (explosive AND (force OR grip OR strength)) OR (force-time)) AND (dynamometer)”. The applied filters were “English”, “Portuguese”, and “Human”. These terms have been adapted for searching in Scopus. 

### 2.4. Selection Process, Data Collection Process, and Data Items

Two authors independently selected the titles, abstracts, and full text of the studies. Discrepancies in selections were discussed until consensus was reached or a third author decided. One author extracted the data, and another author independently verified all extracted data. Any discrepancies were confirmed by a third author.

The extracted data included author, year of publication, title, country, study methods [study design, model, and type of dynamometer; individual positioning for the test (posture and position of the elbow, forearm, and shoulder)], HGS protocol, grip duration, number of repeated tests, duration of the rest period, hand tested, whether encouragement was provided during the assessment, participant characteristics (e.g., age, sex, socioeconomic status), health status, and outcomes (description of indicators and variables used).

### 2.5. Study Risk of Bias Assessment

Two reviewers jointly assessed the quality of this study. Discrepancies in the selections were discussed until a consensus was reached.

Quasi-experimental studies were evaluated using the Risk Of Bias In Non-randomized Studies—of Interventions (ROBINS-I) tool (2016 update) [48], and cross-sectional studies were evaluated using the Newcastle–Ottawa Scale (NOS) tool adapted for cross-sectional design [49,50].

## 3. Results

### 3.1. Study Selection 

The search identified 3296 records and 4 additional records were added by scanning the reference lists of articles included in this article, as well as in systematic reviews and protocols that emerged from the data search. After discarding duplicates (*n* = 367), 2929 records were selected for review of the titles and abstracts, leaving 47 complete articles for eligibility assessment. Of these, 32 complete articles were excluded. The reasons for excluding these studies were as follows: not using a dynamometer (*n* = 1), not assessing HGS (*n* = 10), not evaluating strength as a function of time (*n* = 9), not describing HGS indicators (*n* = 11), and the type of study was a protocol (*n* = 1) (Figure 1).

### 3.2. Study Characteristics 

Table 1 describes an overview of the 15 studies included in this systematic review.

### 3.3. Participants Characteristics

The studies included were published between 1987 and 2023 and the study sample ranged from 10 to 962 participants.

Concerning the location, four studies were conducted in Japan [51,52,53,54], four in the USA [37,40,41,55], two in Australia [56,57], two in Jordan [42,43], two in the UK [38,39] and one in Denmark [44]. Regarding health conditions, seven studies were carried out with healthy participants [41,42,43,44,52,53,54], and four studies included both healthy participants and participants with some health conditions, namely, fractures of the distal radius [38], rheumatoid arthritis [39], muscle complaints [37], and post-stroke [56]. Three studies included only participants with one health condition, including patients with upper limb injuries [55], patients who visited the memory disorders clinic for the first time [51], and community-dwelling adults with upper motor neuron syndrome after acquired brain injury [57]. Finally, one study included master-aged endurance athletes [41].

**Table 1 nutrients-16-01951-t001:** Overview of the characteristics of the included studies.

Study Author (year), Country	Participants	Health Condition	Study Design	Dynamometer Model and Type	Grip Duration	Number of Repeated Tests	Rest Time	Indicators	Study Quality
P Helliwell [39] (1987) UK	Group A: *n* = 20, most of whom were women ^#^, aged 18–30 years Group B: *n*= 30 (*n* = 20 women, *n* = 10 men), 47–90 years Group C: *n*= 46 (*n* = 33 women, *n* = 13 men), 33–77 years	Healthy group and group of patients with rheumatoid arthritis	Cross-sectional	Extensometer torsion dynamometer	4.4 s	2 to 3	NI	Time to the maximum value T-90% Fatigue rate Grip fatigue (%) Release rate Power factor	3/10 ^‡^
S N Chengalur [55] (1990) USA	*n* = 60 (*n* = 30 men, mean age 27.2 years, and *n* = 30 women, 28.5 years)	Upper extremity injuries	Quasi-experimental	Modified Jamar adjusted	5 s	3	1 min	Plateau coefficient of variation	Serious risk ^§^
Y Ikemoto [54] (2006) Japan	*n* = 30 ^#^, mean age of men 21.9 years and mean age of women 21.4 years	Healthy	Cross-sectional *	Digital hand dynamometer (EG-290, Sakai, Japan)	5 s	3	1 min	Time to the maximum value T-90% Inflection point Int0.25 s, Int0.5 s, Int1 s	5/10 ^‡^
S Demura [53] (2008) Japan	*n* = 10 men, 20–26 years	Healthy	Cross-sectional	Digital hand dynamometer (EG-290, Sakai, Japan)	NI	12, 15, 20, and 30 grips per minute	5, 4, 3, and 2 s	Time to 80% maximal voluntary contraction Average integrated area Final force value Inflection Time	6/10 ^‡^
I J Baguley [56] (2010) Australia	*n* = 5 (*n* = 2 men and *n* = 3 women), mean age 54 years	Healthy and post-stroke patients	Pilot	Computerized hand dynamometer (G100 Precision Dynamometer; Biometrics Pty Ltd.; Jamar configuration)	NA	Hand grasp and release during repetitive maximal force generation. 10 cycles	NI	Cycle time Time between cycles Maximum force velocity Minimum force velocity Grip work	Serious risk ^§^
Z D Alkurdi [42] (2010) Jordan	*n* = 20 men students	Healthy	Cross-sectional	Grip force transducers attached to a power laboratory	As long as possible	1	2 min	50% fatigue	4/9 ^‡^
K Watanabe [52] (2011) Japan	*n* = 57 (*n* = 30 healthy young women, mean age 22.3 years, and *n* = 27 healthy older women, mean age 78.5 years)	Healthy	Cross-sectional	Digital hand dynamometer (EG-210, Smedley type; Sakai Co. Ltd., Chiyoda-ku, Tokyo)	6 s	3	3 min	Maximal rate of grip force development Rate of grip force development	5/9 ^‡^
H Barden [57] (2012) Australia	Patients: *n*= 36 ^#^, mean age 50 years. Control participants: *n* = 27 ^#^, mean age 40 years	Upper motor neuron syndrome following acquired brain injury	Quasi-experimental	Jamar-style Biometrics G100 Precision Dynamometer. The raw dynamometer signal was sampled at 400 Hz and amplified through a general-purpose amplifier to a Power-Lab 26	NI	3	NI	Cycle duration Grip work Maximum force velocity Minimum force velocity	Serious risk ^§^
Y Matsui [51] (2014) Japan	*n* = 347 patients (*n* = 205 women and *n* = 142 men), mean age 75 years	Patients who visited the memory disorders clinic for the first time	Cross-sectional	A newly developed dynamometer device for measuring grip strength (made by IMADA, Toyohashi, Japan)	NI	NI	NI	Time to the maximum value Response time	5/9 ^‡^
CE Plant [38] (2015) UK	*n* = 25 patients (*n* = 10 women and *n* = 15 men), mean age 40 years,	Healthy, and patients with a fracture of the distal radius	Quasi-experimental	Tracker Freedom wireless dynamometer (JTECH Medical, Salt Lake City, UT, USA) using the version 5 software	10 s	6	15 s	Grip fatigue	Serious risk ^§^
M LI [37] (2018) USA	*n*= 19 subjects with muscular complaints (*n* = 5 women and *n* = 14 men), mean age 46.6 years; *n* = 18 subjects without muscular complaints (*n* = 4 women and *n* = 14 men), mean age 45.3 years,	Participants with muscular complaints and healthy	Quasi-experimental	Digital hand dynamometer	NI	3	NI	Endurance Fatigue level (%)	Serious risk ^§^
L De Dobbeleer [44] (2019) Denmark	Group A: *n* = 175 men, mean age 57.9 years. Group B: *n* = 100 (*n* = 41 men, mean age 56.6 years; *n* = 59 women, 56.8 years) Group C: *n* = 687 (*n* = 319 men, mean age 50.1 years; *n* = 368 women, mean age 50.0 years)	Healthy	Cross-sectional	JD G100 system, consisting of an adjustable hand grip handle (standard JD configuration) equipped with an in-built compression load cell and connected via a strain gauge amplifier to a computer	Maintain maximum pressure for as long as possible	3	30 s	Fatigue resistance Grip work	6/9 ^‡^
L Klawitter [40] (2020) USA	*n* = 13 (*n* = 7 women and *n* = 6 men), mean age 70.9 years	NI	Cross-sectional	Biopac hand grip dynamometer and Student Lab software (Biopac; Goleta, CA, USA)	10 s; as long as possible	2	30 s	Submaximal control 25% Fatigability index	5/10 ^‡^
S F Almashaqbeh [43] (2022) Jordan	*n* = 100 (*n* = 41 women, mean age 21.8 years, *n* = 59 men, mean age 22.1 years)	Healthy	Cross-sectional ^†^	Grip force transducer connected to the Power Lab unit (AD Instruments corporation).	Maintain maximum pressure for as long as possible	7	5 min	Fatigue resistance 25% Fatigue resistance 50% Fatigue resistance 75%	5/10 ^‡^
L Klawitter [41] (2023) USA	*n* = 31 (*n* = 8 women and *n* = 23 men), mean age 49.1 years	Healthy	Pilot	Biopac hand grip dynamometer and Student Lab software (Biopac; Goleta, CA, USA)	1, 10 s; as long as possible	2	1 min	Rate of hand grip force development Hand grip fatigability index Submaximal hand grip force control (coefficient of variation)	4/9 ^‡^

Abbreviations: NI—no information; NA—not applicable; Int—integrated area. * Study designated as “crossover” by the study authors but classified as cross-sectional by the review team according to the Dictionary of Epidemiology description [58]. ^†^ Study designated as “experimental“ by the study authors but classified as cross-sectional by the review team according to the Dictionary of Epidemiology description [58]. ^‡^ Assessment of study quality using the Newcastle–Ottawa Scale. The studies were evaluated with a total scale score between 9 and 10 points, due to the non-applicability of a criterion. ^§^ Assessment of study quality using the ROBINS-I tool. ^#^ Information regarding the number of male or female participants is absent.

### 3.4. Hand Grip Strength Measurement

HGS was assessed with different types of dynamometers, such as digital dynamometers [37,38,40,41,51,52,53,54,56], dynamometers with grip force transducers coupled to computers [42,43,55], extensometer torsion dynamometers connected to a microprocessor for timed compression analysis [39], dynamometers equipped with a load cell built-in compression sensor and connected via a strain gauge amplifier to a computer [44], or Jamar-style Biometrics G100 precision dynamometers in which the raw dynamometer signal was sampled at 400 Hz and amplified through a general-purpose amplifier to a Power-Lab 26 [57] (Table 1).

Of the fifteen studies analyzed in the present review, only two mentioned the protocol used to measure HGS [38,57], which was the American Society for Hand Therapists Guidelines [59], but in one study, modifications to this original protocol were made [57].

Regarding the positioning of the individual for the dynamometry test (posture and position of the elbow, forearm, and shoulder), HGS was evaluated in different anatomical positions. In two studies, HGS was measured in seven different ways, including adduction of the arm with 180° at the elbow joint, adduction of the arm with 90° forward at the elbow joint with the arm perpendicular to the frontal plane, abduction of the arm with 90° at the shoulder joint and 180° at the elbow, abduction of the arm with 90° at the shoulder joint and 90° at the elbow joint with the forearm perpendicular to the frontal plane, abduction of the arm with 180° at the shoulder joint and 180° at the elbow joint, and abduction of the arm with 180° at the shoulder joint shoulder and 90° at the elbow joint [42,43]. In another study, HGS was measured with participants standing, with their arms parallel to the body [54]. In the study by Barden et al. [57], participants self-selected a static wrist position to perform the grip and release with the elbow and forearm supported. In two studies, participants were seated with their shoulders adducted and in neutral rotation, their forearm in a neutral position, and their elbow flexed at 90 degrees [38,40]. In the study by Watanabe et al., the procedure was the same, but the elbow was fully extended [52]. Matsui et al. [51] only mentioned that grip strength was measured in the sitting position, with the elbows flexed at approximately 90°. In the study by Baguley et al. [56], participants were seated in a chair with armrests or in a wheelchair and the position of the upper limb was standardized with the elbow supported at 90° of flexion, and the forearm positioned in neutral, allowing a wrist extension of 0–30° and ulnar deviation of 0–15°. In the study by Klawitter et al. [41], participants sat with their forearms on a chair, with their wrists in a neutral position just above the end of the chair arm, and with their thumb facing up. In the study by Demura et al. [53], participants performed the HGS sitting in an adjustable ergometric chair with the arm in a sagittal and horizontal position, supported by an armrest with the forearm vertical and the hand in a semi-prone position. In four studies, there was no information about the position in which HGS was measured [37,39,44,55].

In nine of the included studies, HGS was assessed in both hands [37,38,39,40,41,42,55,56,57], three studies assessed HGS in the dominant hand [44,53,54], but three studies did not provide information on the hand used [43,51,52].

The duration of the grip varied between 1 and 10 s, whereas in five studies, participants were instructed to maintain maximum strength for as long as possible [40,41,42,43,44]. Four studies provided no information [37,51,53,57], and in one, the duration of the grip did not apply [56] (Table 1). The duration of the rest period between grips varied between 2 s and 5 min. Five studies provided no information [37,39,51,56,57] (Table 1).

An incentive during the assessment of HGS was provided in four studies [40,41,53,57], while nine studies did not provide information concerning this matter [37,39,42,43,44,51,54,55,56].

### 3.5. Hand Grip Strength Indicators

The hand grip force–time curve indicators identified in this systematic review were grip fatigue; fatigability index; fatigue rate; fatigue resistance 25%, 50%, and 75%; time to 80% maximal voluntary contraction; plateau coefficient of variability; time to a maximum value; T-90% (time until reaching 90% of the maximum peak value); release rate; power factor; grip work; average integrated area; endurance; cycle duration; the time between cycles; maximum force–velocity; minimum force–velocity; rate of grip force development; maximal rate of grip force development; maximal rate of grip force development normalized; final force value; inflection point; integrated area during 0.25 s, 0.5 s and 1 s; submaximal control 25%; and response time.

The hand grip force–time curve indicators identified in this systematic review and their assessment methods are described in Table 2.

To increase knowledge on the utility of using the hand grip force–time curve indicators, the main findings between the several indicators and health, performance, and applicability are summarized in Table 3. 

The findings on the fatigue indicators (grip fatigue (%); fatigability index; fatigue rate; fatigue resistance 25%, 50%, and 75%; and time to 80% maximal voluntary contraction) are summarized in a single parameter in Table 3 called “fatigue”. It is also summarized in one parameter of the rate of grip force, the rate of grip force development (RGFD), the maximal rate of grip force development, and the maximal rate of grip force development normalized.

### 3.6. Risk of Bias in Studies 

The assessment of the quality of the studies included in this systematic review is described in Table 1. In cross-sectional studies evaluated using the Newcastle–Ottawa Scale, the total scale score was nine and ten points, due to the non-applicability of a criterion (non-respondents). Therefore, in studies where the maximum scale was ten points, the quality varied between three and six. In studies where the maximum scale was nine, the quality varied between four and six.

All studies evaluated with the ROBINS-I tool were classified as possessing serious risk bias.

## 4. Discussion

This systematic review identified 23 HGS indicators evaluated in the force–time curve. The most frequently found indicators were related to hand grip fatigue, namely, grip fatigue (%) [37,38,39]; fatigability index [40,41]; fatigue rate [39]; and fatigue resistance 25% [43], 50% [42,43,44], and 75% [43]. Another indicator of fatigue resistance was found, but the authors named it “time to 80% maximal voluntary contraction” [53].

Muscle fatigue can be defined as a reduction in the ability to generate force in a tense situation [45,60]. Several neuromuscular, musculoskeletal, and metabolic factors influence the propensity to fatigue [45,46]. Fatigue is a significant dimension of hand function that is often neglected in patients with upper limb injuries or disorders, although it has the potential to identify hand and wrist impairments [35,61,62]. In addition, increased muscle fatigue may explain the occurrence of fatigue, one of the main characteristics of frailty in the elderly [44]. Fatigability analysis is an additional clinical approach, offering a more accurate way to identify older adults at risk for a rapid decline in physical function. Since the fatigue indicators reviewed in this article are calculated in different ways, as shown in Table 2, it would be relevant in the future to homogenize the measurement of fatigue for it to be possible to compare between subjects.

Grip work and time to maximum value indicators were also among the most evaluated indicators in the analyzed articles. Grip work refers to the functional capacity resulting from the development of a certain level of strength concerning the time that can be maintained, reflecting the total effort produced during the fatigue resistance test [44,63]. The assessment of this indicator is reflected in daily activities that require continuous muscular effort, such as lifting, handling, or transporting objects [63]. 

The time to maximum value indicator may be valuable in patients with rheumatoid arthritis since patients with rheumatoid arthritis take significantly longer to reach maximum grip strength than healthy individuals. It has been suggested that the prolongation of time to maximum adhesion is equivalent to the subjective experience of stiffness in rheumatoid disease [39]. Although these findings may provide some information about the usefulness of assessing time to maximum grip strength, they must be interpreted with caution because this study scored 3 out of 10 in the quality assessment. In the study by Matsui et al. [51], it was found that the response time indicator was significantly related to the total Barthel Index [64], which assesses activities of daily living. 

The rate of grip force development is the most used parameter in evaluating the generation of muscular strength [65]. It is defined as the slope of the force–time curve obtained during the isometric muscle contraction [66,67]. The rate of force development at different times is quantified, although the greatest rate of force development occurs in the first 50–75 ms of muscle contractions [25,68]. The justification for analyzing specific aspects of the force–time curve is based on the relative contribution of the neural and muscular systems over time [25,68].

The rate of grip force development has an important functional significance as it analyzes the fast and strong muscle contractions that are necessary for sports [66] and it appears to be more influenced by aging than maximum HGS [52].

In this review, we identified different ways of calculating the rate of force development and it is important in the future to standardize procedures, that is, to analyze the best formula to calculate the rate of force development so that the indicator can be used as an assessment tool, providing more detailed information about hand grip function.

The submaximal control indicator reflects the ability to control muscle strength during submaximal contractions and is essential for evaluating basic manual dexterity [69]. Furthermore, the submaximal control indicator was negatively correlated with maximum strength, and it could be an indicator that provides more information in the assessment of HGS [41], namely offering information about poor concentration, low neuromuscular control, and reduced neuromuscular function [25].

From the analysis of the included studies, the indicators related to fatigue seem to be useful for assessment in distal radius fracture and rheumatoid arthritis, whereas in rheumatoid arthritis, “time to maximum value” could also provide additional information. Grip work was different among affected upper limb and healthy subjects, and cycle duration and time between cycles could be useful, respectively, for patients with acquired brain injury and post-stroke patients. The maximum force velocity and minimum force velocity can also be useful for post-stroke and acquired brain injury patients. The rate of grip force was related to aging, whereas response time was related to functionality.

Regarding study limitations, as far as we know, there is no information in the scientific literature on some indicators targeted in this review about their potential in evaluating muscle function, namely, the plateau coefficient of variation; release rate; power factor; average integrated area; endurance; cycle duration; the time between cycles; maximum/minimum force–velocity; final force value; inflection time; and integrated area during 0.25, 0.5, and 1 s. It is essential to investigate this relationship in the future and take into consideration the characteristics of the participants. Moreover, there is a dearth of studies conducted among subjects with high levels of physical activity, so the ability of the identified indicators to assess their muscular performance remains to be ascertained. 

Among the studies included, we found that participants assessed HGS with different body postures and anatomical hand positions, which have an impact on the HGS values obtained [70,71], making it very important to homogenize procedures and determine protocols for evaluating revised hand grip indicators. Nevertheless, in the future, the existing HGS protocols can be easily adapted to incorporate the indicators obtained from the force–time curve.

Furthermore, the authors of the included articles used different dynamometers to evaluate the indicators, some of which have not been validated with others that are very old. Although this could have impacted the accuracy of the values, the objective of this review was to identify indicators that can be assessed through hand grip strength, and thus the type of dynamometer was not a reason for exclusion.

Another limitation is that the participants from the studies included in the review have a wide age range, including young adults and older adults. In the future, it will be important to apply force–time curve indicators stratified by age to understand their utility on muscle function assessment in all age groups. 

We highlight that digital dynamometers that provide information on the HGS force–time curve have real potential in the future. They can also provide several indicators that could complement the maximum grip strength indicator, which is very important in evaluating muscle function and therefore people’s health and athletic performance. It has the advantage of being a tool that can be used for primary, secondary, and tertiary intervention purposes. Therefore, future studies conducted to assess muscle function should explore indicators obtained from the force–time curve, in addition to maximal HGS.

This review was a first step towards understanding which indicators can be obtained from the HGS force–time curve, making it essential in the future to analyze the association between force–time curve indicators and muscle mass in health and disease. 

## 5. Conclusions

To the best of our knowledge, this is the first review of hand grip force–time curve indicators. With this review, we describe several indicators in addition to maximum grip strength that can be evaluated through the time–force curve using a digital or adapted dynamometer. These indicators can be useful in evaluating health parameters, such as the decline in physical function, the ability to perform activities of daily living, and neuromuscular control.

In future investigations, it is essential to analyze these indicators, to understand the implications on muscle function, to assess which indicators other than maximum HGS could provide important information, to standardize procedures to evaluate them, and to determine the implications in clinical practice.

## Figures and Tables

**Figure 1 nutrients-16-01951-f001:**
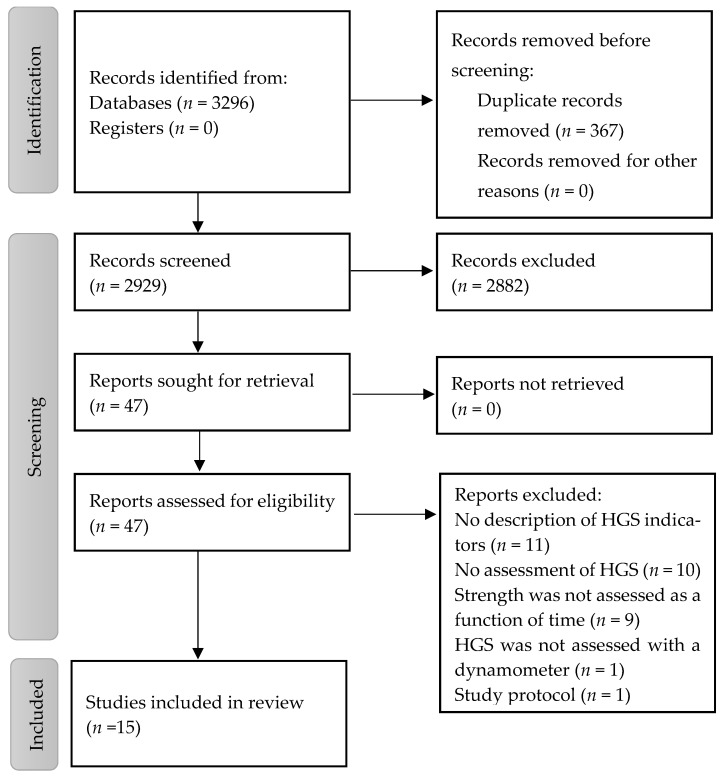
PRISMA flow chart of included studies. Abbreviations: HGS—handgrip strength.

**Table 2 nutrients-16-01951-t002:** Hand grip force–time curve indicators identified in this systematic review and their assessment method.

Indicator	Assessment Method
Grip fatigue (%) [37,38,39]	Percentage of loss of grip from the maximum grip strength to the point of release of the grip. $Fatigue\;\left(\%\right)=\frac{\left(Maximum\;HGS-Final\;HGS\right)100}{Final\;HGS}$
Fatigability index [40,41]	$Fatigue\;\left(\%\right)=1-\left(\frac{Real\;area}{Ideal\;area\;\left(Maximum\;HGS*time\right)}\right)*100\%$
Fatigue rate [39]	Rate of loss of grip from maximum value to the point of release of grip, expressed in newtons per second.
Fatigue resistance 25% [43] Fatigue resistance 50% [42,43,44] Fatigue resistance 75% [43] Time to 80% maximal voluntary contraction [53]	Time required for hand grip strength to decrease to 25%, 50%, 75%, or 80% of its maximum during maximum sustained hand grip effort.
Plateau coefficient of variation [55]	$Coefficient\;of\;variation=\frac{SD\;of\;plateau*100}{Mean\;of\;plateau}$ Plateau region of the force–time curve from the occurrence of 90% of peak to end of the contraction.
Time to maximum value [39,51,54]	The time it takes to reach the maximum.
T-90% [39,54]	Time until reaching 90% of the maximum peak value.
Release rate [39]	Rate of loss of grip from the point of release to the baseline value, expressed in newtons per second.
Power factor [39]	The integral of the grip/time curve, expressed in newton seconds.
Grip work [44,56,57]	The area under the curve is calculated by integrating the actual grip force at each time interval (i.e., 0.01 s) based on measurements recorded during the drop in grip force to 50% of its maximum value. $Grip\;work=\sum_{(Maximum\;HGS-50\%\;of\;maximum\;HGS)\;}HGS*t$ *t*—time interval (at 100 Hz = 0.01 s)
Average integrated area [53]	Average of all strength values during sustained isometric grip and repeated rhythmic grip during the stipulated minutes.
Endurance [37]	Time that participants maintain the grip on the dynamometer for as long as possible in the range of 70% to 90% of maximum HGS.
Cycle duration [56,57]	The duration of time of the cycle from the minimum grip strength until the minimum grip strength is reached again.
Time between cycles [56]	Time that passes between repeated cycles of maximum contraction and relaxation.
Maximum force velocity [56,57] Minimum force velocity [56,57]	The force–velocity curve (force/time) is the speed at which a person changes their isometric force production during the grip and release of the dynamometer. The area above the horizontal zero line represents the gripping phase, which starts at zero and increases until the maximum force–velocity curve, defined as the point at which peak velocity occurs. The release phase is represented below the horizontal zero line, also starting at zero and decreasing to the minimum level, the point at which release is being generated most quickly.
Rate of grip force development (RGFD)	$RGFD=\Delta \;force^{\prime }/\Delta \;time$ Defined as the slope of the force–time curve at intervals of 10 ms up to 250 ms, with each measurement using point 0 as the baseline [52]. The rate of force development is calculated as peak force (kgs) normalized to time (s) [41].
Maximal rate of grip force development (Max RGFD) [52]	max RGFD=Δ force/10 ms Maximum value of the slope of the force–time curve in each 10 ms interval.
Maximal rate of grip force development normalized (Max RGFD) [52]	$Normalized\;max\;RFGD=\frac{Max\;RFGD}{Maximum\;HGS}$
Final force value [53]	The final force value is the force in a defined time (e.g., 360 s).
Inflection point [53,54]	The inflection point of decreasing velocity in the force–time curve distinguishes between a sharp decreasing phase and a nearly stationary phase during repeated rhythmic grips. The time-series-sustained force data were divided into the former and later phases at all combinations and the respective regression lines were calculated. The inflection point was determined at the time from the best-fitting regression lines in the combination of time series data.
Int0.25 s Int0.5 s Int1 s [54]	Integrated area during 0.25, 0.5, and 1 s.
Submaximal control 25% [40,41]	This is calculated as 25% of the maximum HGS. Participants were asked to squeeze the dynamometer and maintain a submaximal target grip strength of 25%. The coefficient of variation was calculated during the intermediate period of 8 s.
Response time [51]	Time from time zero until grip strength begins.

Abbreviations: HGS—handgrip strength; Int—integrated area; SD—standard derivation.

**Table 3 nutrients-16-01951-t003:** Hand grip force–time curve indicators identified in this systematic review and their association with health issues, performance, and potential applicability.

Indicator	Main Findings
Fatigue Grip fatigue (%) [37,38,39] Fatigability index [40,41] Fatigue rate [39] Fatigue resistance 25% [43] Fatigue resistance 50% [42,43,44] Fatigue resistance 75% [43] Time to 80% maximal voluntary contraction [53]	The level of fatigue (%) in a group with carpal tunnel syndrome was similar to that in healthy individuals and there were no significant correlations with the estimated number of motor units [37]. Grip fatigue in patients with distal radius fracture was related to hand and wrist impairment [38]. Individuals with rheumatoid arthritis showed considerably more fatigue than healthy individuals. In this group of patients, fatigue (%) was independent of maximum HGS; however, the fatigue rate in the healthy group and the individuals with rheumatoid arthritis was closely related to HGS [39]. Assessment of fatigability may be an alternative clinical tool, providing a more accurate identifier of older people at risk of a rapid decline in physical function [44].
Plateau coefficient of variation [55]	In upper extremity injuries, the plateau coefficient of variation distinguished sincere effort from fake effort. The sincere trials typically showed a rapid increase in force, reaching a plateau at near-peak force levels where the force remained relatively constant. In contrast, the faking trials also exhibited a rapid initial increase in force but often included an initial “spike” where the subject overshot the intended force application, resulting in an early peak force [55].
Time to maximum value [39,51,54]	The time to maximum grip strength in a group of patients with rheumatoid arthritis was significantly prolonged compared to the healthy group [39]. The time to the maximum value in patients who visited a memory disorders clinic for the first time was not significant in either gender or in the maintenance of elderly patients’ independence in activities of daily living (assessed by the Barthel Index) [51].
T-90% [39,54]	No information.
Release rate [39]	The release rate in the healthy group and group of patients with rheumatoid arthritis was closely related to maximum grip strength and probably did not provide any additional information [39].
Power factor [39]	No information.
Grip work [44,56,57]	In post-stroke patients, the degree of grip work performed by the affected upper limb of the post-stroke groups was related to less than the grip work performed by the unaffected upper limb. These findings appear to be typical of adults with negative upper motor neuron features such as weakness, reduced motor control, and fatigue [56].
Average integrated area [53]	The average integrated area was closely related to the final force values [53].
Endurance [37]	Endurance was slightly affected by carpal tunnel syndrome, suggesting that it is a potentially important addition to motor unit number estimation for the longitudinal follow-up of motor neuron function among veterans with carpal tunnel syndrome [37].
Cycle duration [56,57]	Patients with acquired brain injury demonstrate a slower cycle duration, consistent with negative features of upper motor neuron syndrome [57]. In post-stroke patients, the cycle time was longer in both affected and unaffected limbs compared with healthy subjects. The increased time to complete cycles suggests slower motor recruitment patterns when performing handgrip contraction and relaxation in the post-stroke group [56].
Time between cycles [56]	In post-stroke patients, the time between cycles was longer in both affected and unaffected limbs compared with healthy individuals. A longer time between repeated cycles suggests difficulty in resuming the contraction phase of the task after relaxation, potentially due to slower muscle recruitment, biomechanical changes after the previous contraction phase, and/or motor planning difficulties [56].
Maximum force velocity [56,57] Minimum force velocity [56,57]	In post-stroke patients, the time required to reach maximum and minimum velocities (from the onset of muscle contraction) was slower than in healthy individuals. Therefore, the maximum velocity of force (maximum rate of contraction) was severely impaired, coinciding with negative upper motor neuron features such as weakness, reduced motor control, and fatigue. The post-stroke group also showed a longer time to reach maximum relaxation, that is, reduced minimum force velocity [56]. Patients with acquired brain injury had lower maximum and minimum force velocity values when compared to healthy individuals [57].
Rate of grip force Rate of grip force development (RGFD) [41,52] Maximal rate of grip force development (Max RGFD) [52] Maximal rate of grip force development normalized (Max RGFD) [52]	The rate of force development is frequently used to assess muscle force generation. It has important functional significance, as rapid and strong muscle contractions are required in sports and a high rate of contractile force development exerted during the initial phase of muscle contraction can be very important for successful performance. Age-related decreases in grip force generation capacity have been found for maximal RFFD and normalized maximal RFFD, possibly due to altered muscle contraction capacity [52] RGFD was moderately and positively correlated with maximal grip strength in endurance athletes over 35 years of age [41].
Final force value [53]	The final force values were closely related to the average integrated area [53].
Inflection point [53,54]	Physiological factors such as muscle fiber recruitment and muscle oxygenation related to strength effort differ at pre- and post-inflection points [53]. The inflection point is a useful parameter for dividing force and contraction velocity because it relates to the force–time parameters that evaluate the maximum development phase, but not to the peak value.
Int0.25 s Int0.5 s Int1 s [54]	The areas integrated up to 0.25 and 0.5 s were related to muscle contraction velocity and potentially assess explosive muscle function [54].
Submaximal control 25% [40,41]	Submaximal strength was moderately and negatively correlated with maximal HGS [41].
Response time [51]	Response time was associated with some items related to activities of daily living (assessed by the Barthel Index) [51].

Abbreviations: HGS—handgrip strength; Int—integrated area.

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
