# Peer review of "Hand Grip Force–Time Curve Indicators Evaluated by Dynamometer: A Systematic Review"

_nutrients, 2024, doi:10.3390/nu16121951_

Round 1

Reviewer 1 Report

Comments and Suggestions for Authors

The objective of this systematic review was to identify indicators of the HGS measurement profile based on the force-time curve in addition to maximum grip strength, which is the widely used strength indicator.

The work is well structured and deals with a topic of potential interest for the scientific community.

I only have a few minor suggestions for authors.

In particular, it would be appropriate to specify the acronyms used in the Prsim Flowchart in the relevant caption. The same goes for the other tables.

Furthermore, in consideration of the importance of the topic covered, and the number of articles present in the literature, I believe that the introductory paragraph is too summary and should be expanded.

Author Response

The authors acknowledge the Reviewer's comments and questions and the opportunity to submit an improved and revised version of the manuscript. We would like to thank the reviewers for their valuable comments and suggestions, which have greatly contributed to improving our work.

Please find our answers to the reviewers below, in blue text. The changes that were made to the manuscript are highlighted in yellow.

REVIEWER #1

  1. “In particular, it would be appropriate to specify the acronyms used in the Prism Flowchart in the relevant caption. The same goes for the other tables.”

R: We have included this information in Prisma Flowchart: HGS, and also in Table 2: HGS, SD, INT.

  1. “Furthermore, in consideration of the importance of the topic covered, and the number of articles present in the literature, I believe that the introductory paragraph is too summary and should be expanded.”

R: The introductory paragraph of “Introduction” was expanded.

Reviewer 2 Report

Comments and Suggestions for Authors

General Comments: This paper systematically reviews hand grip strength (HGS) indicators beyond maximum HGS using force-time curves evaluated by dynamometer. It identifies various indicators such as grip fatigue, rate of force development, and fatigue resistance, aiming to enhance muscle function assessment. The focus of the review adds value to the many fields that may use HGS for assessing varying aspects of human performance and function across the lifespan. However, several aspects of the manuscript warrant further attention.

The writing can be improved. In the introduction and discussion several paragraphs were very short and did not fully develop ideas presented.

A forward citation search was not performed of the included studies. This could have resulted in studies that met the eligibility criteria being missed. Suggest that a forward citation search is performed.

The purpose(s) of the systematic review could be more meaningful. My interpretation of the purpose, based on the last paragraph of the introduction, is to identify which HGS measures, other than just strength, are reported in the literature. Taking this 1 step further to also focus on which specific HGS indicators are found to be associated with various health issues and/or performance adds value to the review.

The discussion was difficult to follow and some of what was presented would fit better in the introduction to provide rationale for the review (see earlier comments). It seems that other than strength there were 2 general times of HGS indicators: 1) rate of force development and 2) fatigue. Again, I would urge the authors to add more discussion around which specific HGS indicators were associated with various health issues and performance. A schematic or table could be useful for the discussion. Adding paragraphs on implications for practitioners and researchers (e.g., future research directions) would strengthen the discussion.

It is unclear why the conclusion wasn’t a single paragraph and was presented as 4 very short paragraphs.

Specific Comments

Paragraph 1: in addition too various adverse health HGS holds value as a fitness and recovery measure in higher performing populations. Particularly some of the force-time metrics focused on in the review. Suggest adding text to reflect that HGS applies across a spectrum of functioning levels.

Paragraphs 2 and 3 can be combined into a single paragraph.

A greater explanation of the force-time indicators needs to be incorporated in the introduction with some mention of the neurophysiology processes that would influence the indicators. Specifically some of the indicators in table 2 should be introduced in the introduction.

Line 36: need space before [7, 8]

Line 59-60: Some examples of neuromuscular related disorders that would influence some of the HGS indicators can be added here.

Line 69: I believe ‘systematize’ should be ‘synthesize’

Line 81-82: Can simply state that all populations were included regardless of health status.

Did 2 authors also assess the risk of bias independently then discuss until consensus was reached?

Participant characteristics – please add the total number of participants in the included studies and number of males and females.

Line 284 – need space before [64]

Comments on the Quality of English Language

The writing can be improved. In the introduction and discussion several paragraphs were very short and did not fully develop ideas presented.

Author Response

The authors acknowledge the Reviewer's comments and questions and the opportunity to submit an improved and revised version of the manuscript. We would like to thank the reviewers for their valuable comments and suggestions, which have greatly contributed to improving our work.

Please find our answers to the reviewers below, in blue text. The changes that were made to the manuscript are highlighted in yellow.

REVIEWER #2

  1. “The writing can be improved. In the introduction and discussion several paragraphs were very short and did not fully develop ideas presented.” 

R: To overcome this issue, the “Introduction” and “Discussion” sections were improved.

  1. “A forward citation search was not performed of the included studies. This could have resulted in studies that met the eligibility criteria being missed. Suggest that a forward citation search is performed.”

R: A forward citation search was conducted as stated in the Materials and Methods section: “The studies were identified by searching on August 25, 2023, in electronic databases and scanning reference lists of the articles included in this article and in systematic reviews or protocols that emerged in the search for data.”

And in the Results section: “The search identified 3296 records and 4 additional records were added by scanning the reference lists of articles included in this article,…”

  1. “The purpose(s) of the systematic review could be more meaningful. My interpretation of the purpose, based on the last paragraph of the introduction, is to identify which HGS measures, other than just strength, are reported in the literature. Taking this 1 step further to also focus on which specific HGS indicators are found to be associated with various health issues and/or performance adds value to the review.”

R: This relevant information was added to the “Abstract”, and “Introduction”, and the identified indicators associated with health or muscular function were further developed in the new Table 3.

  1. “The discussion was difficult to follow and some of what was presented would fit better in the introduction to provide rationale for the review (see earlier comments). It seems that other than strength there were 2 general times of HGS indicators: 1) rate of force development and 2) fatigue. Again, I would urge the authors to add more discussion around which specific HGS indicators were associated with various health issues and performance. A schematic or table could be useful for the discussion. Adding paragraphs on implications for practitioners and researchers (e.g., future research directions) would strengthen the discussion.”

R: The “Introduction” and “Discussion” sections were improved, namely, future implications were added to the Discussion.

Moreover, a new table (Table 3) containing information about the association of HGS indicators with health status and performance was inserted.

  1. “It is unclear why the conclusion wasn’t a single paragraph and was presented as 4 very short paragraphs.”

R:  The conclusion was reformulated as suggested:

“To the best of our knowledge, this is the first review of hand grip force-time curve indicators. With this review, we describe several indicators in addition to maximum grip strength that can be evaluated through the time-force curve using a digital or adapted dynamometer. These indicators can be useful in evaluating health parameters, such as the decline in physical function, the ability to perform activities of daily living, and neuromuscular control.

In future investigations, it is essential to analyze these indicators, to understand the implications on muscle function, to assess which indicators other than maximum HGS could provide important information, to standardize procedures to evaluate them, and to determine the implications in clinical practice.”

Reviewer #2 Specific Comments:

  1. “Paragraph 1: in addition to various adverse health HGS holds value as a fitness and recovery measure in higher performing populations. Particularly some of the force-time metrics focused on in the review. Suggest adding text to reflect that HGS applies across a spectrum of functioning levels.”

R: To address this issue, the text was rephrased and completed: “It is a diagnostic tool to identify sarcopenia, undernutrition, physical frailty, and the recovery from these diseases and processes [24-26]. Moreover, HGS is a commonly used parameter to assess physical performance and fitness [2].”

  1. “Paragraphs 2 and 3 can be combined into a single paragraph.”

R: Paragraphs 2 and 3 were combined into a single paragraph.

  1. “A greater explanation of the force-time indicators needs to be incorporated in the introduction with some mention of the neurophysiology processes that would influence the indicators. Specifically some of the indicators in table 2 should be introduced in the introduction.”

R: We added it in the “Introduction” section.

  1. “Line 36: need space before [7, 8]”.

R:  Corrected.

  1. “Line 59-60: Some examples of neuromuscular related disorders that would influence some of the HGS indicators can be added here.”

R:  We added it in the “Introduction” section.

  1. “Line 69: I believe ‘systematize’ should be ‘synthesize’.”

R:  Corrected.

  1. “Line 81-82: Can simply state that all populations were included regardless of health status.”

R:  This information was added to the “Methods” section.

  1. “Did 2 authors also assess the risk of bias independently then discuss until consensus was reached?”

R:  The two reviewers jointly assessed the quality of the articles. This information was added to the “Methods” section.

  1. “Participant characteristics – please add the total number of participants in the included studies and number of males and females.”

R:  Table 1 was changed accordingly.

  1. “Line 284 – need space before [64]”

R: Corrected.

Reviewer 2# Comments on the Quality of English Language

“The writing can be improved. In the introduction and discussion several paragraphs were very short and did not fully develop ideas presented.” 

R:  Besides the edition of the “Introduction” and “Discussion” sections, the entire manuscript was edited by a native English-speaking revisor.

Round 2

Reviewer 2 Report

Comments and Suggestions for Authors

The author's revised manuscript is much better in my opinion. Particularly, table 3 is valuable for researchers and practitioners in a variety of areas. I have no further edits.

Comments on the Quality of English Language

Writing is generally sound. Another review for grammar, spelling and punctuation is recommended, which is my standard advice.